# Carbon Redistribution and Microstructural Evolution Study during Two-Stage Quenching and Partitioning Process of High-Strength Steels by Modeling

**DOI:** 10.3390/ma11112302

**Published:** 2018-11-16

**Authors:** Yilin Wang, Huicheng Geng, Bin Zhu, Zijian Wang, Yisheng Zhang

**Affiliations:** State Key Lab of Materials Processing and Die and Mould Technology, Huazhong University of Science and Technology, Wuhan 430074, China; wangyilin@hust.edu.cn (Y.W.); ghc@hust.edu.cn (H.G.); zhubin26@hust.edu.cn (B.Z.); zhangys@mail.hust.edu.cn (Y.Z.)

**Keywords:** advanced high-strength steel, two-stage Q-P, interface migration modeling, carbon partitioning, retained austenite

## Abstract

The application of the quenching and partitioning (Q-P) process on advanced high-strength steels improves part ductility significantly with little decrease in strength. Moreover, the mechanical properties of high-strength steels can be further enhanced by the stepping-quenching-partitioning (S-Q-P) process. In this study, a two-stage quenching and partitioning (two-stage Q-P) process originating from the S-Q-P process of an advanced high-strength steel 30CrMnSi2Nb was analyzed by the simulation method, which consisted of two quenching processes and two partitioning processes. The carbon redistribution, interface migration, and phase transition during the two-stage Q-P process were investigated with different temperatures and partitioning times. The final microstructure of the material formed after the two-stage Q-P process was studied, as well as the volume fraction of the retained austenite. The simulation results indicate that a special microstructure can be obtained by appropriate parameters of the two-stage Q-P process. A mixed microstructure, characterized by alternating distribution of low carbon martensite laths, small-sized low-carbon martensite plates, retained austenite and high-carbon martensite plates, can be obtained. In addition, a peak value of the volume fraction of the stable retained austenite after the final quenching is obtained with proper partitioning time.

## 1. Introduction

Recently, the quenching and partitioning (Q-P) process has been applied to advanced high-strength steels to obtain austenite (*γ*) + martensite (*α*) multiphase microstructures, which exhibit high strength and excellent ductility [1,2,3,4]. In the Q-P process, the quenching of austenite between the martensite start temperature (*M_s_*) and the martensite finish temperature (*M_f_*) results in martensite laths and untransformed austenite. Then, in the partitioning step, the steel is held at the same or higher temperature to promote carbon diffusion from supersaturated martensite to untransformed austenite. Thus, the process improves the carbon concentration in austenite, and the austenite retains greater stability at room temperature. Consequently, in the material, martensite confers high strength, and the retained austenite provides good ductility due to the TRIP effect [5,6,7,8].

The Q-P process is regarded as a means to obtain chemically stabilized austenite. Several Q-P process variants have been proposed to further extend the concept [2,9]. The quenching, partitioning and tempering (Q-P-T) process emphasizes the effects of alloy carbides, and results in higher strength and elongation for some alloy steels [10,11]. The stepping-quenching-partitioning (S-Q-P) process involves multiple quenching and partitioning steps [12,13]. A type of mixed microstructure was formed after the treatment of the S-Q-P process, as follows: a substrate consisting of low-carbon martensite laths with characteristics of high ductility and strength; some retained austenite distributed in long thin strips, which improved the toughness of the steel; and lower-carbon twinned martensite and martensite with high dislocation density, contributing to the strength improvement of the steel. Finally, an excellent combination of mechanical properties was obtained in a medium-carbon steel by using the S-Q-P process [12].

To better understand the mechanism of the Q-P process, several models have been proposed to describe the partitioning process of carbon or other alloying elements [14,15,16,17,18]. Speer et al. [14] proposed a Constrain Carbon Equilibrium (CCE) model to describe carbon partitioning from martensite to austenite. In the CCE model, it is assumed that, during the partitioning process, competitive reactions are fully suppressed, the *α*/*γ* interface is stationary, and carbon’s chemical potential in martensite and austenite is equal at the interface. Santofimia et al. [15,16] also put forward a model which depicted interface migration and carbon diffusion quantitatively, similarly by assuming the same chemical potential of carbon in austenite and martensite at the interface. Incorporating carbide precipitation and bainite formation into simulation models can capture a more realistic carbon redistribution process [19,20,21,22]. However, in this paper, we aimed to study the carbon redistribution and microstructure evolution during the two-stage Q-P process.

The two-stage Q-P process originates from the S-Q-P process, and involves two quenching processes, in which the second quenching temperature (T_Q2_) is lower than the first quenching temperature (T_Q1_), while both of them are between the *M_s_* and *M_f_* temperatures. Each quenching process is followed by a partitioning process either at or above the initial quenching temperature. Previous studies have mainly been focused on the traditional Q-P process [23,24,25,26]. However, the two-stage Q-P process has rarely been explored, despite exhibiting part performance superior to the traditional process. Furthermore, due to the difficulty of observing the microstructure of the material after the two-stage Q-P process, it is meaningful to investigate the process by simulation, which can present the carbon distribution and the evolution of the microstructure clearly.

In this paper, the microstructural evolution and carbon redistribution of high-strength steel 30CrMnSi2Nb during the two-stage Q-P process were investigated by using Santofimia’s model. Additionally, the effects of the first and second quenching temperature were analyzed, as well as the influence of partitioning time on interface migration and carbon distribution. In addition, the optimal process parameters were determined for two-stage Q-P process, with which relatively more austenite and an ideal alternately distributed microstructure can be obtained. Finally, corresponding XRD experiments were implemented to verify the simulation.

## 2. Materials and Methods

The chemical composition of the high-strength steel 30CrMnSi2Nb, with a nominal thickness of 1.5 mm, used in this study is listed in Table 1. A certain amount of carbon ensures the formation of stable retained austenite and high-carbon twinned martensite. Silicon can suppress the precipitation of cementite and facilitates the stabilization of austenite during the partitioning reaction [1,27].

The two-stage Q-P process includes two quenching processes, and each process is followed by a partitioning process. While the one-stage Q-P process includes only one partitioning process. The schematic diagrams of the two-stage and one-stage Q-P processes are shown in Figure 1. The two-stage Q-P process consists of the following procedures: (a) quenching the fully austenitized steel to the first quenching temperature (T_Q1_); (b) holding the steel at the first partitioning temperature (T_P1_) to achieve incomplete partitioning; (c) quenching the steel again to the second quenching temperature (T_Q2_); (d) conducting another incomplete partitioning at the second partitioning temperature (T_P2_); (e) finally quenching the steel to room temperature. The one-stage Q-P process consists of the following procedures: (a) quenching the fully austenitized steel to the first quenching temperature (T_Q_); (b) holding the steel at the first partitioning temperature (T_P_) to achieve incomplete partitioning; (c) finally quenching the steel to room temperature. To investigate the effects of the parameters on the process, T_Q1_ was set to 325 °C and 350 °C, respectively; T_P1_ was set to 425 °C, and the first partitioning time was 10 s; T_Q2_ was set to the temperature of 150 °C, 200 °C, and 250 °C, respectively; T_P2_ was 350 °C, and the second partitioning time was 10 s. The microstructures and carbon distributions at different stages of the two-stage Q-P process are displayed in Figure 2.

XRD measurements were conducted to determine the volume fraction of retained austenite and austenite carbon content. Cu Kα radiation was used, and a 2*θ* range from 40° to 101° was scanned using a step size of 0.04°. The integrated intensities of the (200), (220) and (311) austenite peaks and the (200), (211) and (220) ferrite peaks were measured to calculate the volume fraction of retained austenite [9]. In addition, the retained austenite carbon content was determined from the austenite lattice parameter using the method given in Ref. [9].

Two basic elements are involved in the partitioning processes of the two-stage Q-P process: carbon redistribution in *α*, *γ*, the *α*/*γ* interface, and the *γ*/*α* interface; and interface migration [23]. The kinetic process of carbon diffusion was modeled by means of the one-dimensional finite difference method (FDM). During the simulation process, the precipitation of carbides, as well as the decomposition of bainite formations by austenite, was neglected. Meanwhile, only the partitioning of carbon was calculated, without consideration of the effects of other alloying elements.

### 2.1. Carbon Diffusion at the Interface

Previous studies have indicated that it is the chemical potential gradient, rather than the concentration gradient, that actually influences carbon diffusion. Thus, at the beginning of the first partitioning process, carbon diffuses from martensite of higher carbon chemical potential to austenite of lower carbon chemical potential, despite the equal carbon concentration in martensite and untransformed austenite. Moreover, carbon diffusion will not stop until equal carbon chemical potential is reached. Meanwhile, the chemical potential of carbon in martensite and austenite is assumed to be equal at the interface in order to guarantee the continuity of the chemical potential of carbon across the interface. In addition, this relationship can be expressed by carbon concentration as [15]:(1)$$x_{\mathrm{int}}^{\alpha }=x_{\mathrm{int}}^{\gamma }\exp\left[\frac{A+BT+\left(C+DT\right)x_{\mathrm{int}}^{\gamma }}{RT}\right]$$
where $x_{\mathrm{int}}^{\alpha }$ and $x_{\mathrm{int}}^{\gamma }$ are the mole fraction of carbon in *α* and *γ* phases, respectively. *R* and *T* denote the gas constant and absolute temperature, respectively. A, B, C, and D are constants related to the chemical composition of both phases and can be obtained by thermodynamic software, as shown in Table 2.

### 2.2. Interface Migration

It is assumed that the carbon chemical potential at the interface of martensite and austenite is equal, and the effects of substitutional elements are ignored; the driving force for interface migration arises from the difference of chemical potential of iron at the interface. For modeling purposes, martensite is considered to have a carbon supersaturated body-centered cubic (bbc) structure, whereas austenite is the face-centered cubic (fcc) phase. The model considers the same chemical potential of carbon of bcc and fcc in the bcc-fcc interface because of the high atomic mobility of interstitial carbon [15,23]. In this work, this condition has been represented in terms of carbon concentration by the use of Equation (1). The relation between the interface migration velocity v and the driving force ΔG can be expressed as [23]:(2)$$v={\left(V_{m}\right)}^{-1}M\Delta G$$
where *V_m_* is the molar volume of iron, *M* is the interface mobility, and $M=3.36\times 10^{-18}$ m^4^·J^−1^·s^−1^.

Since the effects of alloying elements were neglected except for carbon, the driving force ΔG can be calculated by:(3)$$\Delta G=x_{Fe}^{\alpha }\left(\mu_{Fe}^{\gamma }-\mu_{Fe}^{\alpha }\right)+x_{C}^{\alpha }\left(\mu_{C}^{\gamma }-\mu_{C}^{\alpha }\right)$$
where $x_{Fe}^{\alpha }$ and $x_{C}^{\alpha }$ are the concentrations of iron and carbon in phase *α* respectively, and $x_{Fe}^{\alpha }=1-x_{C}^{\alpha }$. $\mu_{C}^{\gamma }\left(\mu_{Fe}^{\gamma }\right)$ and $\mu_{C}^{\alpha }\left(\mu_{Fe}^{\alpha }\right)$ refer to the chemical potential of carbon (iron) at the interface in *γ* and *α*, respectively. As $\mu_{C}^{\gamma }=\mu_{C}^{\alpha }$ and $x_{C}^{\alpha }$ is tiny, the equation of driving force can be simplified as:
(4)$$\Delta G=\mu_{Fe}^{\gamma }-\mu_{Fe}^{\alpha }$$

Under the condition $\mu_{C}^{\gamma }=\mu_{C}^{\alpha }$, when $x_{C}^{\gamma }$ reaches the equilibrium concentration $x_{C}^{\gamma -eq}$, $\mu_{Fe}^{\gamma }$ is equal to $\mu_{Fe}^{\alpha }$. Then a linear relationship between $\mu_{Fe}^{\gamma }-\mu_{Fe}^{\alpha }$ and $x_{C}^{\gamma }-x_{C}^{\gamma -eq}$ can be reasonably assumed. Thus, ΔG can be described as:(5)$$\Delta G=k\left(x_{C}^{\gamma -eq}-x_{C}^{\gamma }\right)$$
where *k* is a constant of proportionality, and k=1021 J·wt.%^−1^·mol^−1^.

After Equation (5) is substituted into Equation (2), the interface migration velocity *v* is expressed as:(6)$$v={\left(V_{m}\right)}^{-1}M\cdot k\left(x_{C}^{\gamma -eq}-x_{C}^{\gamma }\right)$$

### 2.3. Martensite Formation in Quenching Processes

Martensitic transformation occurs in the first, second and the final quenching process. In the simulation, it is considered that the second martensite (M_2_), formed in the second quenching process, is located in the central region of the retained untransformed austenite after the first partitioning process. In addition, in the final quenching process, the third martensite (M_3_) is generated in the middle region of austenite, featuring a lower carbon concentration. In addition, the volume fraction of martensite is calculated by the integration of the martensite generated in every grid of the untransformed austenite.

### 2.4. Interaction between Carbon Diffusion and Interface Migration

In the simulation, carbon diffusion and interface migration take place at the same time. The motion of interfaces in a microstructure is a result of the repositioning of atoms from lattice positions of one grain to projected lattice positions in a neighboring grain. At a given temperature, the equilibrium concentrations of carbon in fcc and bcc are given by the meta-stable equilibrium phase diagram, excluding carbide formation. If the carbon concentrations in the interface are different from the equilibrium values, the phases will experience a driving pressure for a phase transformation towards the equilibrium phase composition. This local driving pressure is experience at the interface and results in an interface velocity [15,23]. In addition, the solubility of carbon in austenite is much higher than that in martensite. When the interface moves from martensite to austenite, carbon will diffuse from the moving interface to the untransformed austenite near the interface. In addition, when the interface moves from austenite to martensite, carbon will diffuse from the untransformed martensite to the moving interface. Thus, carbon partitioning at the interface can be described by the following equation [28]:(7)$$-D_{C}^{\alpha }{\frac{dc^{\alpha }}{dz}|}_{\mathrm{int}}+v\left(c_{\mathrm{int}}^{\gamma }-c_{\mathrm{int}}^{\alpha }\right)=-D_{C}^{\gamma }{\frac{dc^{\gamma }}{dz}|}_{\mathrm{int}}$$
where $c_{\mathrm{int}}^{\gamma }$ and $c_{\mathrm{int}}^{\alpha }$ are the mole fraction of carbon in austenite and martensite at the interface, respectively.

### 2.5. Carbon Diffusion in Martensite and Austenite

The diffusion of carbon in martensite and austenite is modeled by Fick’s second law of diffusion:(8)$$\frac{\partial C^{i}}{\partial t}=\frac{\partial }{\partial x}\left(D_{C}^{i}\frac{\partial C^{i}}{\partial x}\right)$$
where *i* refers to martensite or austenite, and $D_{C}^{i}$ is the diffusion coefficient of carbon. The diffusion coefficients of carbon in austenite and martensite phases can be calculated by [28]:(9)$$D_{C}^{\gamma }=2\times 10^{-5}\exp\left(\frac{-140000}{RT}\right)$$
(10)$$D_{C}^{\alpha }=2\times 10^{-6}\exp\left(-\frac{10115}{T}\right)\times \exp\left\{0.5898\left[1+\frac{2}{\pi }\arctan\left(1.4985-\frac{15309}{T}\right)\right]\right\}$$
where $D_{C}^{\alpha }$ and $D_{C}^{\gamma }$ refer to diffusion coefficients of carbon in *α* and *γ*, respectively, both of which are evaluated in $m^{2}/s$. In addition, T is the absolute temperature.

### 2.6. Simulation Conditions

The process parameters involved in the simulation were introduced above. Since carbon content is symmetric to the center of martensite and untransformed austenite after the first quenching process, half the thickness of the martensite and austenite was chosen to be the calculation domain. In the simulation, the initial thickness of martensite (after first quenching) is set to 0.2 μm, which is applicable for steel with 0.28 wt.% C [16]. In addition, the thickness of the untransformed austenite film (after first quenching) can be obtained by its volume fraction. Using the Koistinen-Marburger equation [29,30], the volume fraction of austenite is computed by:(11)$$f_{A}^{i}=\exp\left(-\alpha =0.011\left(M_{s}^{i}-T_{Q}^{i}\right)\right)=\frac{d_{\gamma }^{i}}{d_{\alpha }^{i}+d_{\gamma }^{i}}$$
where *i* refers to the first, second or final quenching. $f_{A}^{i}$ is the volume fraction of untransformed austenite to the original austenite in the *i*th quenching. α is a constant, and α=0.011 K^−1^. $T_{Q}^{i}$ corresponds to the *i*th quenching temperature. $d_{\alpha }^{i}$ and $d_{\gamma }^{i}$ are the thicknesses of untransformed austenite and martensite generated in the *i*th quenching process, respectively. $M_{s}^{i}$ (°C) refers to the martensite start temperature of austenite before the *i*th quenching process, and it can be calculated separately in every austenite grid by the following equation [30]:(12)$$M_{s}^{i}=539-423w_{c}-30.4w_{Mn}-7.5w_{Si}+30w_{Al}$$
where $w_{k}$ is the concentration of alloying element *k* in wt.%.

Please note that the simulation procedure is simplified by assuming concentrations of substitutional elements (*j*) as the nominal composition. This K-M equation is applicable if carbon is homogenously distributed in the austenite. However, quenching of specimens may take place before complete carbon homogenization inside austenite grains.

To analyze the unstable austenite with inhomogeneous carbon distribution, the unstable austenite is divided into sub-fractions with homogenous composition. This means the sum of the volume fractions j of unstable austenite with constant carbon concentration ($f^{j}$) is equal to the total fraction of unstable austenite ($f_{UnSA}$) as follows:(13)$$f_{UnSA}=\sum f^{j}$$

According to the K-M equation, the volume fraction of second martensite (M_2_) transformed from unstable austenite can be calculated as follows:(14)$$f_{M_{2}}=\sum_{j}f^{j}\left\{1-\exp\left(-\alpha (M_{s}^{j}-T)\right)\right\}$$

The parameter $M_{s}^{j}$ depends on the chemical composition according to:(15)$$M_{s}^{j}=539-423w_{c}^{j}-30.4w_{Mn}-7.5w_{Si}+30w_{Al}$$
where $w_{c}^{j}$ (wt.%) is the carbon concentration in the segment *j* of unstable austenite. The total volume fraction of stable austenite (*f_RA_*) is given by *f_RA_* = 1 − *f_M_*_1_ − *f_M_*_2_. Then the thickness of stable austenite (*f_RA_*) and second martensite (M_2_) can be calculated using Equation (11) after the second quenching, as shown in Figure 2c.

This calculation method was also used in the paper by F. Hajyakbary et al. [19] to calculate the volume fraction of second martensite M_2_ after the final quenching in the Q-P process. In addition, the position of the newly formed martensite is set to the place where the original austenite reaches the lowest carbon concentration.

The model was implemented in ABAQUS, with the application of user subroutine USDFLD. To better understand the effects of two-stage Q-P process, the numerical simulation of the one-stage Q-P process was also conducted for comparison. Similar to the two-step Q-P process, the one-stage Q-P process includes a quenching process between *M_s_* and *M_f_*, and a partitioning process at a higher temperature.

## 3. Results and Discussion

### 3.1. One-Stage Q-P Process

During the simulation of the one-stage Q-P process for high-strength steel 30CrMnSi2Nb, the first quenching temperature T_Q1_ was set to 250 °C, 275 °C, 300 °C, 325 °C, and 350 °C, respectively. The first partitioning temperature T_P1_ was set to 425 °C, and the partitioning time was 500 s.

#### 3.1.1. Carbon Redistribution during the First Partitioning Process in the One-Stage Q-P Process

Figure 3 shows the carbon redistribution during partitioning at 425 °C for 500 s after quenching to 300 °C. At the beginning of the partitioning process, the carbon content at the interface exhibits a sharp increase in austenite and a sharp decrease in martensite. The carbon content in austenite at the interface reaches a peak value of 4.46 wt.% at 0.01 s. This can be explained by the low carbon solubility in martensite, which results in the diffusion of carbon from supersaturated martensite to untransformed austenite. Moreover, the low diffusion coefficient of carbon in austenite led to the enrichment of carbon in austenite near the interface. Because the carbon concentration in austenite near the interface has exceeded the equilibrium concentration $x_{C}^{\gamma -eq}$, *α/γ* interface migrates from austenite to martensite. The carbon diffusion from the boundary to the central area of austenite, as well as the formation of new austenite through interface migration, continues, consuming the carbon that diffuses from martensite. As a result, carbon concentration declines in austenite at the interface. In this way, the interface migration velocity soon drops to zero and rises in the opposite direction. When the partitioning time reaches 1 s, most of the carbon in the martensite has been transferred to the austenite. As the carbon content in the austenite decreases below the equilibrium concentration, the interface starts to move to the austenite. With the interface movement, the carbon diffuses continuously from the interface to the untransformed austenite, which contributes to the increase of carbon content in austenite. Meanwhile, the carbon content in martensite also increases slightly for equal chemical potentials of carbon at the interface. Carbon in austenite is homogenized at the equilibrium concentration by about 100 s, at which point the interface stops moving.

#### 3.1.2. Retained Austenite after the Final Quenching in the One-Stage Q-P Process

In the one-stage Q-P process, the local volume fraction of the retained austenite after final quenching is shown in Figure 4. With different T_Q1_, different boundary positions for the calculation domain can be obtained. When T_Q1_ increases and much untransformed austenite remains after the first quenching process, the carbon in austenite is not enough to maintain the stability of austenite at room temperature, thus decreasing the fraction of stable retained austenite. Since there exists a carbon concentration gradient in austenite after partitioning for 20 s, the local volume fraction of retained austenite gradually declines from the *α/γ* interface to the middle of the austenite.

Figure 5a shows the evolution of the volume fraction of austenite in the partitioning process, as well as the retained austenite after quenching to room temperature. It is found that there are two peak values of the volume fraction of retained austenite during partitioning. Furthermore, after reaching the second peak value, the volume fractions of retained austenite and untransformed austenite decrease as the partitioning time increases. Finally, the volume fraction reaches a fixed value. Specifically speaking, at the beginning of the partitioning process, the high carbon content of the austenite near the interface causes the first peak of the volume fraction of retained austenite. As illustrated previously, the *α*/*γ* interface moves to the austenite soon after partitioning begins, resulting in an increase in carbon content in the austenite. Furthermore, the stability of the austenite will increase too. When *M_s_* for every grid in austenite drops below room temperature, the martensite transformation terminates, thus creating the second peak. Therefore, it is possible to achieve the greatest austenite retention by using an appropriate partitioning time. Figure 5b shows the maximum acquirable volume fraction of retained austenite with different quenching temperatures and appropriate partitioning times. The maximum volume fraction of retained austenite changes little with different quenching temperatures, while the partitioning time required to reach the maximum increases with the increase in quenching temperature. This is because higher first quenching temperatures give rise to thicker untransformed austenite. The austenite is layered in the martensite matrix, and its thickness varies in response to different conditions, such as differing partitioning times and partitioning temperatures [31]. The layered structure can be also extended and applied in a 3D microstructure, as was observed by D. D. Knijf et al. [31] and F. Wang et al. [12].

#### 3.1.3. Uniformity of Carbon Distribution in Austenite during the First Partitioning Process in the One-Stage Q-P Process

To evaluate the uniformity of carbon distribution in austenite, an equation similar to the standard deviation is proposed:(16)$$D=\sqrt{\frac{1}{n}\sum_{i=1}^{n}{\left(\frac{c_{i}-\bar{c}}{\bar{c}}\right)}^{2}}$$
where *D* refers to an index for carbon distribution uniformity. $\bar{c}$ is the average concentration of carbon in austenite, and $c_{i}$ is the carbon concentration of the *i*th grid across the calculation domain of austenite. The carbon concentration will indeed gradually reach the equilibrium value in modeling. In this paper, 3.1 wt.% was set as the criterion value to determine whether the carbon concentration in austenite was uniform.

Figure 6a shows the evolution of index *D* over partitioning time with different first quenching temperatures. As the partitioning time increases, the carbon concentration in the untransformed austenite reaches a more uniform state faster and with lower first quenching temperature. Since lower first quenching temperature brings about a thinner film of untransformed austenite, the carbon diffuses quickly and distributes evenly in the austenite. As shown in Figure 6b, the partitioning time when the carbon concentration in each grid of austenite reaches 3.1 wt.%, rises, with quenching temperature increasing. For similar reasons to those mentioned above, higher quenching temperature results in thicker untransformed austenite.

### 3.2. Two-Stage Q-P Process

In the two-stage Q-P process, T_Q1_ was set to 325 °C and 350 °C; T_P1_ was 425 °C, and the first partitioning time was 10 s; T_Q2_ was set to 150 °C, 200 °C, and 250 °C, respectively. T_P2_ was 350 °C, and the second partitioning time was 100 s. Higher T_Q1_ leads to thicker untransformed austenite after the first quenching process, which benefits subsequent heat treatment. Similar to the discussion of the one-stage Q-P process, carbon redistribution during the second partitioning process is analyzed, as well as the volume fraction of retained austenite.

#### 3.2.1. Carbon Redistribution during the Second Partitioning Process in the Two-Stage Q-P Process

After a first partitioning of 10 s, most of the carbon in martensite is transferred to austenite, and the *α*/*γ* interface moves a certain distance to the austenite. During the second quenching, new second martensite (M_2_) will form inside the austenite if the *M_s_* in the local region of the austenite is higher than the quenching temperature T_Q2_. Then, it is assumed that the M_2_ sheet will be generated in the center of austenite.

Figure 7 shows the carbon redistribution and interface migration during the second partitioning at 350 °C for 100 s. There are two interfaces in the calculation domain. As for the *α/γ* interface generated in the first quenching process, the interface keeps migrating towards austenite with the increase in partitioning time due to the carbon concentration in austenite being lower than the equilibrium concentration. Furthermore, at the *γ/α* interface generated in the second quenching process, owing to the high carbon concentration but small volume of M_2_, the carbon concentration in austenite near the *γ/α* interface increases rapidly. However, the carbon concentration does not exceed the equilibrium concentration; thus, the driving force for the migration of the *γ/α* interface directs it towards austenite all the time. With both of the interfaces constantly moving towards austenite, carbon keeps diffusing from the interfaces to the nearby austenite. In addition, in the austenite, the carbon also diffuses from the boundary to the inside part. As the second partitioning time increases, the carbon concentration in the austenite grows symmetrically to the center of the austenite. Eventually, the carbon concentration in austenite continues increasing until the equilibrium concentration is reached.

#### 3.2.2. Retained Austenite after Final Quenching in the Two-Stage Q-P Process

Figure 8a,b shows the local volume fraction of retained austenite after the final quenching when T_Q1_ is 325 °C and 350 °C, respectively. Similar to the one-stage Q-P process, during the two-stage Q-P process, a higher second quenching temperature (T_Q2_) also leads to the reduction of the volume fraction of stable retained austenite at room temperature, due to the thicker untransformed austenite with higher T_Q2_. Meanwhile, when T_Q2_ is relatively higher, the concentration gradient of carbon in austenite results in a decrease in the volume fraction of retained austenite from the *α*/*γ* and *γ*/*α* interfaces to the middle of austenite. In addition, with a higher first quenching temperature, thicker M_2_ may form after the second quenching process, which is harmful for microstructure refinement.

Similar to Figure 5a, Figure 9a shows the evolution of the volume fraction of untransformed austenite during the second partitioning process and retained austenite after the final quenching. In addition, there also exists a peak value of the volume fraction of retained austenite. Figure 9b shows that the maximum volume fraction of the retained austenite increases as T_Q2_ rises, while the partitioning time corresponding to the maximum value also increases. Additionally, the simulation results also indicate that the maximum volume fraction of retained austenite with T_Q1_ of 325 °C is larger than the maximum obtained when T_Q1_ is 350 °C.

As previously analyzed, the second partitioning time strongly affects the microstructure and the volume fraction of stable retained austenite. Figure 10a–c depicts the final microstructure when the second partitioning time is 10 s, 20 s, and 30 s, respectively. Obviously, shorter second partitioning times result in the formation of new high-carbon martensite sheet M_3_, while longer second partitioning times give rise to a higher volume fraction of stable retained austenite. Consequently, the microstructure can be associated with the content of retained austenite through the use of an appropriate second partitioning time. In this study, the second partitioning time of 20 s makes a good combination of suitable microstructure and higher content of retained austenite. Finally, through the two-stage Q-P process, the microstructure with alternating distribution of small-sized low-carbon martensite plates, retained austenite, and high-carbon martensite sheets, in addition to the low-carbon martensite laths, can be obtained, which facilitates microstructure refinement. In addition, these special microstructural characteristics were observed experimentally in a medium carbon steel [12].

#### 3.2.3. Uniformity of Carbon Distribution in Austenite during the Second Partitioning in the Two-Stage Q-P Process

The uniformity of carbon distribution in austenite is evaluated by the same index *D* proposed above during the second partitioning in the two-stage Q-P process. Figure 11a,b shows the evolution of index *D* with the increase in second partitioning time for different T_Q1_ of 325 °C and 350 °C, respectively. As displayed in the profiles of index *D*, the uniformity of carbon distribution deteriorates rapidly in the initial stage of the second partitioning process, and then improves quickly as the partitioning time increases. Moreover, the uniformity of carbon distribution worsens with an increase in T_Q2_. Considering that higher T_Q2_ results in more untransformed austenite, carbon diffusion requires a longer partitioning time.

Figure 11c indicates that a longer second partitioning time is required for the carbon concentration in each grid of austenite to reach 3.1 wt.%, despite the increase in T_Q1_ or T_Q2_. With quenching temperature increasing, the volume fraction of untransformed austenite increases too, and the interface has to migrate further to achieve a carbon concentration of 3.1 wt.%. This is why more second partitioning time is needed.

### 3.3. Interface Migration and Migration Velocity

#### 3.3.1. One-Stage Q-P Process

Figure 12a shows the evolution of interface migration and migration velocity during the first partitioning process with different T_Q1_. The interface migration distance becomes longer with the increase of T_Q1_. This is because higher T_Q1_ results in more untransformed austenite. In this way, it takes more time for carbon concentration to reach the equilibrium concentration $x_{C}^{\gamma -eq}$. Figure 12b,c shows interface migration and migration velocity with T_Q1_ of 300 °C, and the detailed evolution in the first second, respectively. It is found that there exists a small migration to martensite at the beginning of the partitioning process, which is soon followed by the *α*/*γ* interface migration to austenite.

Considering the effect of varying partitioning temperature on interface migration, a simulation with increasing partitioning temperature was carried out, in which T_Q1_ was set to 300 °C, T_P1_ rose from 300 °C to 425 °C in 160 s according to the formula $T_{P1}=125{\left(t/160\right)}^{1/3}+300$, and partitioning temperature was held for 500 s. The evolution of interface migration and the velocity in this condition is displayed in Figure 12d. Compared with Figure 12b, greater interface migration is noticed. Since the partitioning temperature is lower at the beginning of partitioning, carbon diffusion in austenite is suppressed. As a result, the high carbon concentration near the interface takes more time to decrease below the equilibrium concentration, and the interface migration to martensite in the actual partitioning process is larger than that in partitioning at a stationary temperature. However, the final interface migration distance turns out equal in both conditions.

#### 3.3.2. Two-Stage Q-P process

In the two-stage Q-P process, before the carbon concentration in austenite reaches the equilibrium concentration, the interface keeps moving, driven by the chemical potential difference of iron between the martensite and austenite at the interface. Figure 13 shows the interface migration and the migration velocity during both partitioning processes in the two-stage Q-P process. The *α*/*γ* interface generated in the first quenching process and *γ*/*α* interface generated in the second quenching process keep migrating to the austenite all the time, except for at the very beginning of partitioning. Furthermore, in the second partitioning process, the interface migration velocity of the two interfaces tends to coincide and eventually turns to zero simultaneously. During the second partitioning process, with respect to the interface near which the austenite has higher carbon content, the interface migration velocity towards austenite is low. Accordingly, less carbon diffuses from the interface to the austenite near the interface. Meanwhile, the higher carbon concentration triggers more carbon diffusion from the austenite near the interface to the inside austenite. However, in the case of the interface with a lower carbon content in austenite, things work in the opposite way. As a consequence, the carbon concentration in austenite at both interfaces tends to coincide, along with the velocity of the two interfaces. Finally, when the carbon concentration reaches the equilibrium concentration, interface migration stops.

### 3.4. XRD Tests to Obtain the Volume Fraction of Austenite and Carbon Content in Austenite

It’s really necessary to carry out corresponding experiments to prove the credibility of the simulation. Therefore, XRD tests were performed. The volume fraction of retained austenite and the carbon content in austenite under different treatment conditions are experimentally obtained. The XRD patterns of the specimens treated with different quenching temperature T_Q1_ and T_Q2_ are shown in Figure 14. The specimens are treated using the two-stage Q-P process. Firstly, quenching the fully austenitized specimens to T_Q1_; secondly, holding the specimens at 425 °C for 10 s; thirdly, quenching the specimens again to T_Q2_; fourthly, partitioning the specimens at 350 °C for 10 s; finally, quenching the steel to room temperature. To investigate the effects of the parameters on the process, T_Q1_ is set to 325 °C and 350 °C, respectively; T_Q2_ is set to 150 °C, 200 °C, and 250 °C, respectively.

The volume fraction of retained austenite after two-stage Q-P process is calculated using the following equation [9]:(17)$$V_{\gamma }=1.4I_{\gamma }/\left(I_{\alpha }+1.4I_{\gamma }\right)$$
where $V_{\gamma }$ refers to the volume fraction of retained austenite. $I_{\gamma }$ and $I_{\alpha }$ refer to the average integral intensity of the (200), (220) and (311) austenite peaks and the average integral intensity of the (200), (211) ferrite peaks.

The carbon concentration in the retained austenite is calculated by [9]:(18)$$C^{\gamma }=\left(a^{\gamma }-3.5467\right)/0.046$$
where $C^{\gamma }$ refers to the austenite carbon concentration in weight percent, and $a^{\gamma }$ refers to the lattice parameter of the austenite in Angstroms. The lattice parameter is calculated by:(19)$$a^{\gamma }=\frac{\lambda \sqrt{h^{2}+k^{2}+l^{2}}}{2\sin\theta_{hkl}}$$
where λ, (*h k l*), and $\theta_{hkl}$ are the wavelength of the radiation, the three Miller indices of a plane, and the Bragg angle, respectively.

Experimental and predicted results for the volume fraction of retained austenite after the final quenching as a function of the quenching temperature are shown in Figure 15a. It is observed that the evolution of the experimental volume fraction of retained austenite fits well with the predicted result. In both cases, the volume fraction increases first and then decreases as second partitioning temperature increases. The carbon contents in the austenite obtained by experiments and simulations are shown in Figure 15b. When the second partitioning temperature is low, the experimental results differ with the simulation ones. This may be due to the carbide precipitation in the martensite, which reduces the carbon content during partitioning.

## 4. Conclusions

The microstructural evolution and carbon redistribution in the one-stage Q-P process and two-stage Q-P process were assessed by modeling. Moreover, the interface migration, uniformity of carbon in austenite and retained austenite after the final quenching were also considered in the simulation. The main conclusions in this paper are summarized as follows:The two-stage Q-P process involves two quenching processes, with the second quenching temperature being lower than the first one, and both of which are between *M_s_* and *M_f_*, with each quenching process being followed by an incomplete partitioning process. The refined microstructure with low-carbon martensite laths, small-sized low-carbon martensite plates, retained austenite, and high-carbon martensite plates can be obtained using the two-stage Q-P process.In both the one-stage Q-P process and the two-stage Q-P process, the peak value of the volume fraction of stable retained austenite can be detected as the partitioning time increases. For example, a peak value of 25.5% can be obtained in the one-stage Q-P process when T_Q1_ = 300 °C, T_P1_ = 425 °C and t_P1_ = 30 s, while a peak volume fraction of 23.0% can be obtained in the two-stage Q-P process when T_Q1_ = 325 °C, T_P1_ = 425 °C and t_P1_ = 10 s, T_Q2_ = 250 °C, T_P2_ = 350 °C and t_P2_ = 30 s. Although the peak value of the volume fraction of retained austenite may be less in the two-stage Q-P process, the peak value is still considerable.In the two-stage Q-P process, a combination of suitable refined microstructure and higher retained austenite content can be achieved by using appropriate quenching temperatures and partitioning times.Partitioning temperature has great influence on interface migration at the beginning of partitioning process. With lower partitioning temperature, the carbon concentration in the austenite near the interface takes a longer time to drop below the equilibrium concentration, intensifying the tendency of interface migration to the martensite. Different heating processes in the partitioning process affect interface migration, but will not change the final conditions in partitioning process.During the second partitioning process in the two-stage Q-P process, as the carbon concentration in the austenite grows symmetrically to the middle of austenite, the interface migration velocity of the two interfaces tends to coincide in the second partitioning process.

## Figures and Tables

**Figure 1 materials-11-02302-f001:**
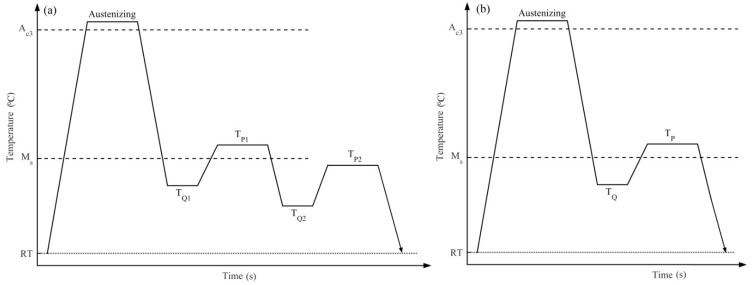
Schematic diagrams of the (**a**) two-stage and (**b**) one-stage Q-P processes.

**Figure 2 materials-11-02302-f002:**
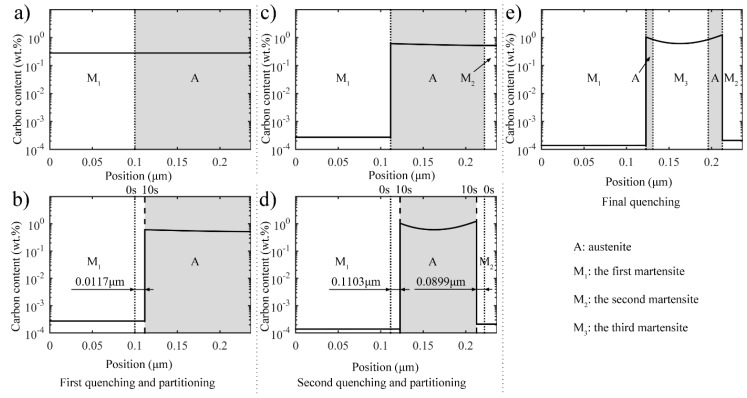
Microstructure and carbon diffusion in different stages of the two-stage Q-P process: (**a**) quenching the steel to the first quenching temperature 325 °C; (**b**) holding the steel at 425 °C for 10 s to achieve incomplete partitioning; (**c**) quenching the steel to the second quenching temperature 250 °C; (**d**) conducting another incomplete partitioning at 350 °C for 10 s; (**e**) quenching the steel to room temperature. The shadow represents austenite, and the distance in the figure marks the interface migration during both partitioning processes.

**Figure 3 materials-11-02302-f003:**
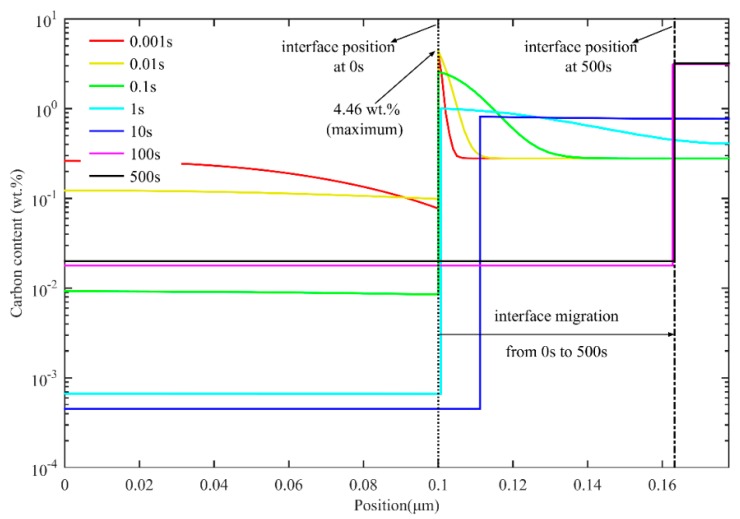
Simulation of carbon redistribution during partitioning at 425 °C for 500 s after quenching to 300 °C (one-stage Q-P process).

**Figure 4 materials-11-02302-f004:**
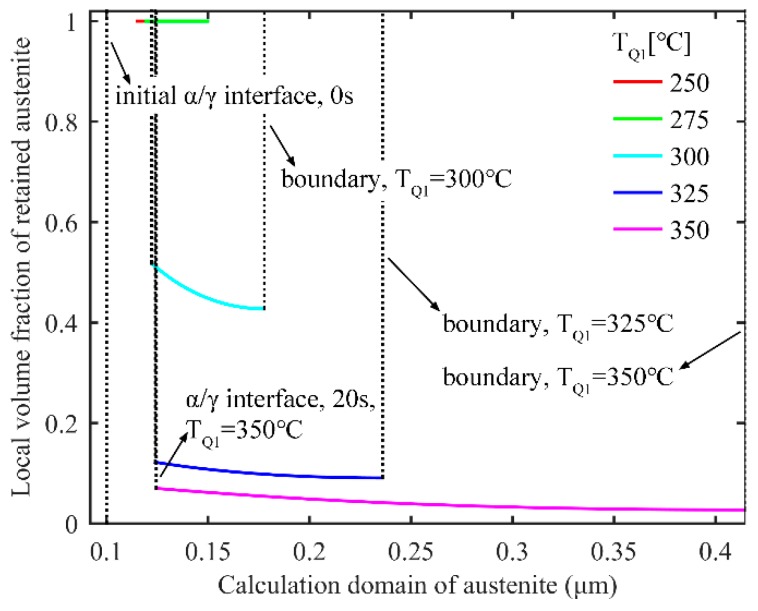
Local volume fraction of retained austenite after final quenching in the one-stage Q-P process. The partitioning temperature and time are 425 °C and 20 s, respectively.

**Figure 5 materials-11-02302-f005:**
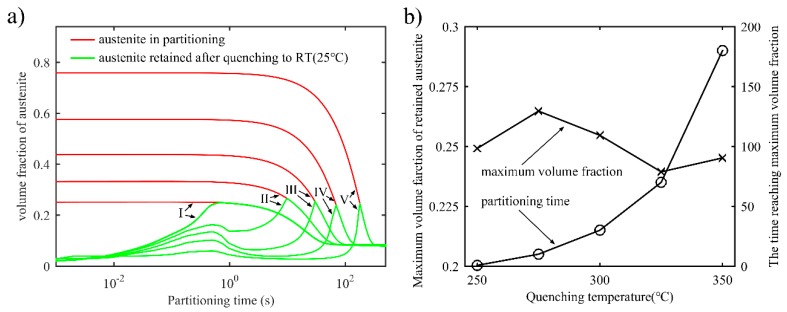
(**a**) Volume fraction of austenite in the partitioning process and retained austenite after quenching to room temperature as a function of partitioning time during the one-stage Q-P process. The partitioning temperature and time are 425 °C and 500 s, respectively, and the first quenching temperature is: I: T_Q1_ = 250 °C; II: T_Q1_ = 275 °C; III: T_Q1_ = 300 °C; IV: T_Q1_ = 325 °C; V: T_Q1_ = 350 °C; (**b**) Maximum volume fraction of retained austenite and the corresponding partitioning time for reaching the maximum value at different partitioning temperatures.

**Figure 6 materials-11-02302-f006:**
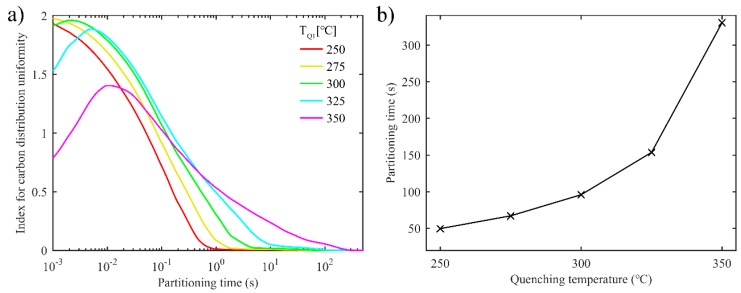
Uniformity of carbon distribution in austenite in the one-stage Q-P process. (**a**) The index for carbon distribution uniformity versus partitioning time; (**b**) The partitioning time when the carbon concentration reaches 3.1 wt.% in each grid of austenite versus first quenching temperature.

**Figure 7 materials-11-02302-f007:**
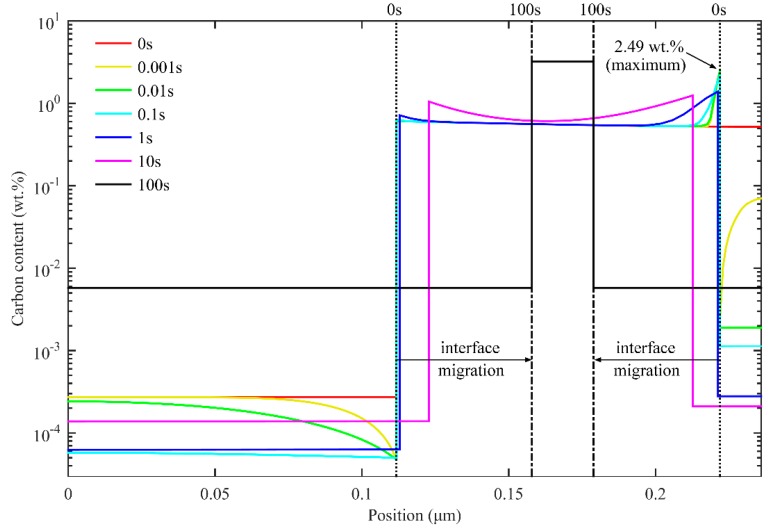
Carbon redistribution during the second partitioning at 350 °C for 100 s after the process of first quenching to 325 °C, first partitioning at 425 °C for 10 s, and second quenching to 250 °C (two-stage Q-P process).

**Figure 8 materials-11-02302-f008:**
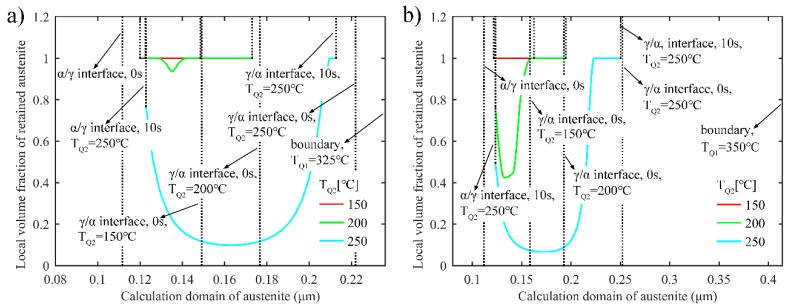
Local volume fraction of retained austenite after the final quenching in the two-stage Q-P process. (**a**) T_Q1_ = 325 °C, T_P1_ = 425 °C, first partitioning time is 10 s, T_P2_ = 350 °C, second partitioning time is 10 s; (**b**) T_Q1_ = 350 °C, T_P1_ = 425 °C, second partitioning time is 10 s, T_P2_ = 350 °C, second partitioning time is 10 s.

**Figure 9 materials-11-02302-f009:**
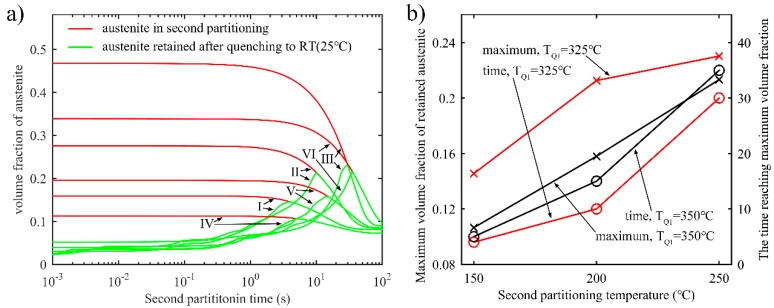
(**a**) Volume fraction of austenite in the second partitioning process and retained austenite after quenching to room temperature as a function of the second partitioning time during the two-stage Q-P process. T_P1_ = 425 °C, the first partitioning time is 10 s, T_P2_ = 350 °C and the second partitioning time is 100 s. The quenching temperature is: I: T_Q1_ = 325 °C, T_Q2_ = 150 °C; II: T_Q1_ = 325 °C, T_Q2_ = 200 °C; III: T_Q1_ = 325 °C, T_Q2_ = 250 °C; IV: T_Q1_ = 350 °C, T_Q2_ = 150 °C; V: T_Q1_ = 350 °C, T_Q2_ = 200 °C; VI: T_Q1_ = 350 °C, T_Q2_ = 250 °C; (**b**) Maximum volume fraction of retained austenite and the corresponding partitioning time for reaching the maximum value at different second partitioning temperature.

**Figure 10 materials-11-02302-f010:**
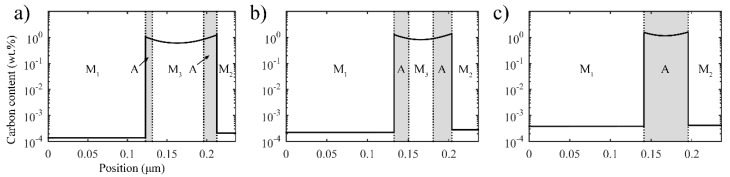
The final microstructure after the two-stage Q-P process corresponding to different second partitioning times. T_Q1_ = 325 °C, T_P1_ = 425 °C and the first partitioning time is 10 s, T_Q2_ = 250 °C, T_P2_ = 350 °C. In addition, the second partitioning time is: (**a**) 10 s; (**b**) 20 s and (**c**) 30 s.

**Figure 11 materials-11-02302-f011:**
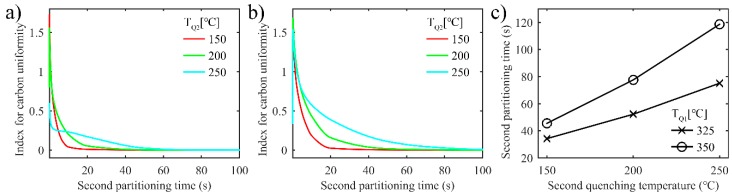
Uniformity of carbon distribution in austenite in the two-stage Q-P process. (**a**) The index for carbon distribution uniformity versus second partitioning time. T_Q1_ = 325 °C, T_P1_ = 425 °C and the partitioning time is 10 s; T_P2_ = 350 °C and the partitioning time is 100 s; (**b**) The index for carbon distribution uniformity versus second partitioning time. T_Q1_ = 350 °C, T_P1_ = 425 °C and the partitioning time is 10 s; T_P2_ = 350 °C and the partitioning time is 100 s; (**c**) The second partitioning time when carbon concentration reaches 3.1 wt.% in each grid of austenite versus second quenching temperature.

**Figure 12 materials-11-02302-f012:**
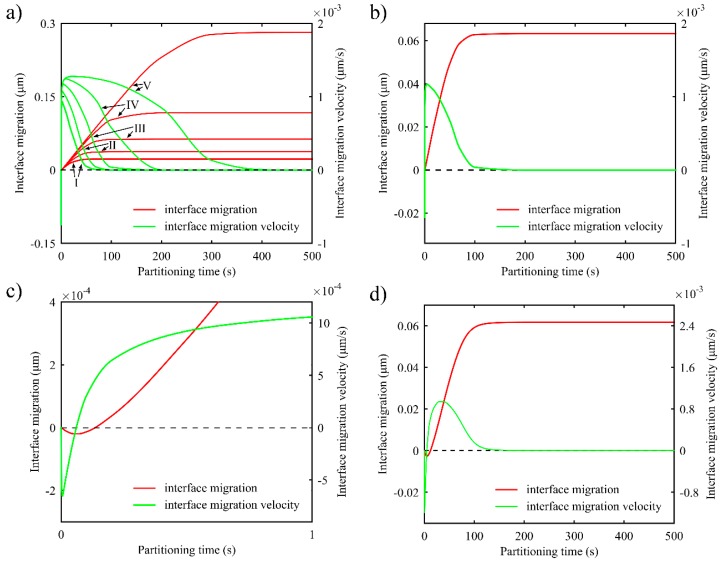
Evolution of interface migration and interface migration velocity during the first partitioning process (two-stage Q-P process). The positive direction of the interface migration points from martensite to austenite. (**a**) The evolution during partitioning at 425 °C for 500 s after quenching to different first quenching temperature: I: T_Q1_ = 250 °C; II: T_Q1_ = 275 °C; III: T_Q1_ = 300 °C; IV: T_Q1_ = 325 °C; V: T_Q1_ = 350 °C; (**b**) The evolution during partitioning at 425 °C for 500 s when T_Q1_ = 300 °C; (**c**) Detailed evolution in the first second when T_Q1_ = 300 °C; (**d**) The evolution when partitioning temperature rises from 300 °C to 425 °C in 160 s according to the formula $T_{P1}=125{\left(t/160\right)}^{1/3}+300$, the partitioning temperature is kept for 500 s, and T_Q1_ = 300 °C.

**Figure 13 materials-11-02302-f013:**
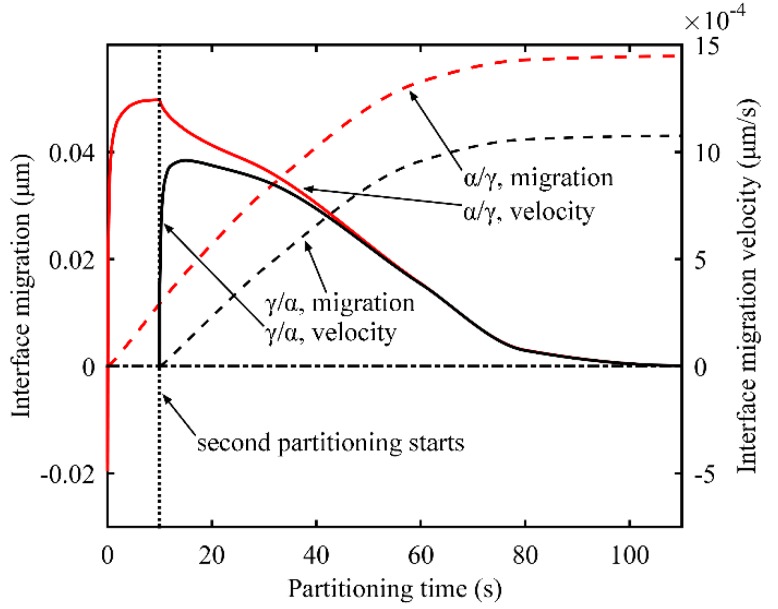
Evolution of interface migration and interface migration velocity during the first and second partitioning (two-stage Q-P process). The positive direction of the interface migration points from martensite to austenite. The parameters for the process are: T_Q1_ = 325 °C, T_P1_ = 425 °C and the partitioning time is 10 s; T_Q2_ = 250 °C, T_P2_ = 350 °C and partitioning time is 100 s.

**Figure 14 materials-11-02302-f014:**
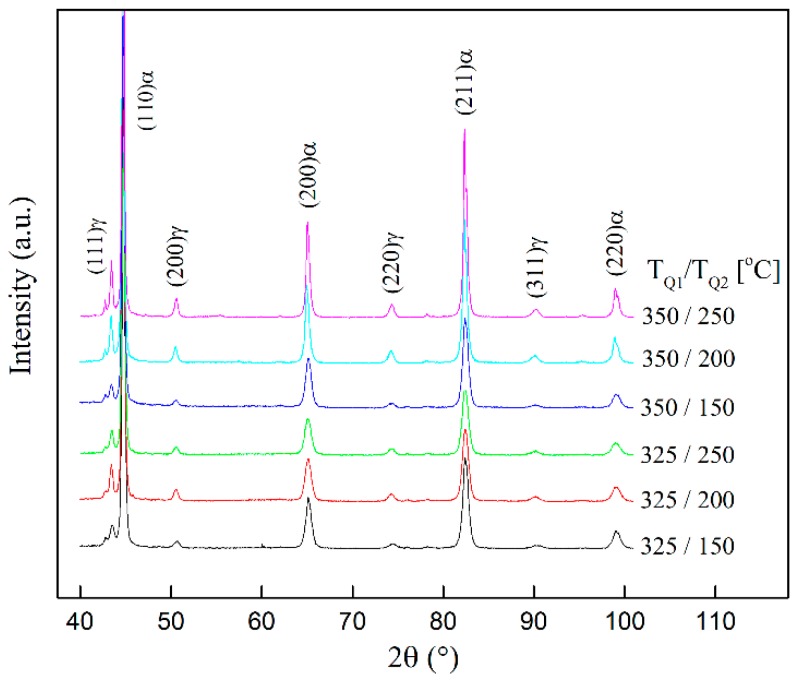
X-ray diffraction patterns of specimens treated with different quenching temperature of T_Q1_ and T_Q2_. T_P1_ = 425 °C, and the first partitioning time is 10 s; T_P2_ = 350 °C, and the second partitioning time is 10 s.

**Figure 15 materials-11-02302-f015:**
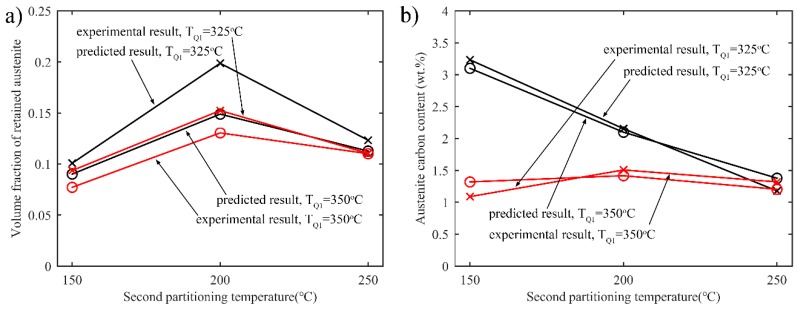
Comparison of the experimental and simulation results for retained austenite content and carbon content in austenite with various partitioning temperatures T_Q1_ and T_Q2_. T_P1_ = 425 °C. The first partitioning time is 10 s; T_P2_ = 350 °C, and the second partitioning time is 10 s. (**a**) Volume fraction of retained austenite; (**b**) Carbon content in austenite.

**Table 1 materials-11-02302-t001:** Chemical composition of high-strength steel 30CrMnSi2Nb (wt.%).

C	Si	Mn	P	S	Cr	Nb
0.28	1.60	1.10	0.012	0.005	0.99	0.029

**Table 2 materials-11-02302-t002:** Constants for Equation (1) in modeling.

A (J·mol^−1^)	B (J·mol^−1^·K^−1^)	C (J·mol^−1^·wt.%^−1^)	D (J·mol^−1^·K^−1^·wt.%^−1^)
−90566	60.5	9776	−5.6

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
