# Peer review of "Carbon Redistribution and Microstructural Evolution Study during Two-Stage Quenching and Partitioning Process of High-Strength Steels by Modeling"

_materials, 2018, doi:10.3390/ma11112302_

Round 1

Reviewer 1 Report

Manuscript ID: materials-386485

(Carbon redistribution and microstructural evolution study during two-stage quenching and partitioning process of high strength steels by modeling), Materials journal

- Review comment

General Comment:

Thank you for an excellent paper on a topic that will become increasingly important as engineers strive to utilize ever higher performance advanced high strength steels (AHSSs) with high strength and improved ductility for demanding automotive and other applications. I enjoyed reading this article. The article is very well written and represents a valuable contribution in the field of the the microstructural evolution and carbon redistribution of high strength steel 6930CrMnSi2Nb during two-stage quenching and partitioning (Q-P) process using Santofimia’s model and the simulation. I can say that I learned a few things about the effect of the application of two-stage Q-P process on advanced high strength steels in order to improve part ductility significantly with little decrease in strength. This is a comprehensive article regarding the both applied simulations of the one-stage Q-P process, two-stage Q-P process, comparison of both processes, optimal process parameters and XRD measurements which were conducted in order to determine the volume fraction of retained austenite and austenite carbon content in steel. The simulation approach is valid and appropriate. Nevertheless, I made several comments related to the minor, but necessary corrections and suggestions in the "Introduction section", "Materials and Methods section", and "Results and Discussion section" (see "Necessary corrections and suggestions" below - Reviewer comment No. 1-16), which I hope the authors can and should address before publication. This additional analysis will greatly contribute to this excellent paper.

 Necessary suggestions and corrections:

The following comments (Reviewer comments No. 1-16) and corrections detailed below should be addressed by the authors. These additional corrections will greatly contribute to this excellent paper:

(1) 1. Introduction section, lines 74-75: "Eventually, the corresponding experiments were implemented to verify the simulation."

Reviewer comment No. 1: What type of experiment is suitable in order to verify the simulation?

(2) 2. Materials and Methods, lines 123-125: "A, B, C, and D are constants related with the chemical composition of both phases and can be obtained by thermodynamic software such as Thermo-calc."

Reviewer comment - correction No. 2: Commercial software name should be deleted and Table 2 text added): "A, B, C, and D are constants related with the chemical composition of both phases and can be obtained by thermodynamic software, Table 2."

(3) 2. Materials and Methods, lines 128-130: "It is assumed that the carbon chemical potential at the interface of martensite and austenite is equal and the effects of substitutional elements are ignored, the driving force for interface migration arises from the difference of chemical potential of iron at the interface."

Reviewer comment No. 3: Some additional explanations for these claims should be added (to cite here – add references here).

(4) 2. Materials and Methods, line 152: "In the simulation, carbon diffusion and interface migration take place at the same time."

Reviewer comment No. 4: An additional explanation for this claims should be added (to cite here – add references here).

(5) 2. Materials and Methods, lines 202-204: "Similar to the two-step Q-P process, the one-stage Q-P process includes a quenching process between Ms and Mf, and a partitioning process at a higher temperature."

Reviewer comment No. 5: Schematic diagram for the one-stage Q-P process should be also added here or at Figure 1 with explanations and comparisons.

(6) 3. Results and Discussion, line 210: "3.1.1 Carbon redistribution during the first partitioning process."

Reviewer comment - correction No. 6: "3.1.1 Carbon redistribution during the first partitioning process in one-stage Q-P process."

(7) 3. Results and Discussion, line 232: "Figure 3. Simulation of carbon redistribution during partitioning at 425 °C for 500 s after quenching to 300 °C."

Reviewer comment - correction No. 7: "Figure 3. Simulation of carbon redistribution during partitioning at 425 °C for 500 s after quenching to 300 °C (one-stage Q-P process)."

(8) 3. Results and Discussion, lines 264-265: "Figure 5. a) Volume fraction of austenite in partitioning process and retained austenite after quenching to room temperature as a function of partitioning time."

Reviewer comment - correction No. 8: "Figure 5. a) Volume fraction of austenite in partitioning process and retained austenite after quenching to room temperature as a function of partitioning time during one-stage Q-P process."

(9) 3. Results and Discussion, line 270: "3.1.3 Uniformity of carbon distribution in austenite during the first partitioning process"

Reviewer comment - correction No. 9: "3.1.3 Uniformity of carbon distribution in austenite during the first partitioning process in one-stage Q-P process."

(10) 3. Results and Discussion, line 287: "Figure 6. Uniformity of carbon distribution in austenite."

Reviewer comment - correction No. 10: "Figure 6. Uniformity of carbon distribution in austenite in one-stage Q-P process."

(11) 3. Results and Discussion, line 297: "3.2.1 Carbon redistribution during the second partitioning process"

Reviewer comment - correction No. 11: "3.2.1 Carbon redistribution during the second partitioning process in two-stage Q-P process"

(12) 3. Results and Discussion, lines 317-318: "Figure 7. Carbon redistribution during the second partitioning at 350 °C for 100 s after the process of first quenching to 325 °C, first partitioning at 425 °C for 10 s, and second quenching to 250 °C."

Reviewer comment - correction No. 12: "Figure 7. Carbon redistribution during the second partitioning at 350 °C for 100 s after the process of first quenching to 325 °C, first partitioning at 425 °C for 10 s, and second quenching to 250 °C (two-stage Q-P process)."

(13) 3. Results and Discussion, lines 341-342: "Figure 9. a) Volume fraction of austenite in the second partitioning process and retained austenite after quenching to room temperature as a function of the second partitioning time."

Reviewer comment - correction No. 12: "Figure 9. a) Volume fraction of austenite in the second partitioning process and retained austenite after quenching to room temperature as a function of the second partitioning time during two-stage Q-P process."

(14) 3. Results and Discussion, line 364: "3.2.3 Uniformity of carbon distribution in austenite during the second partitioning"

Reviewer comment - correction No. 13: "3.2.3 Uniformity of carbon distribution in austenite during the second partitioning in two-stage Q-P process"

(15) 3. Results and Discussion, line 379: "Figure 11. Uniformity of carbon distribution in austenite."

Reviewer comment - correction No. 14: "Figure 11. Uniformity of carbon distribution in austenite in two-stage Q-P process."

(16) 3. Results and Discussion, lines 405-406: "Figure 12. Evolution of interface migration and interface migration velocity during the first partitioning process."

Reviewer comment - correction No. 15: "Figure 12. Evolution of interface migration and interface migration velocity during the first partitioning process (two-stage Q-P process)."

(17) 3. Results and Discussion, lines 430-431: "Figure 13. Evolution of interface migration and interface migration velocity during the first and second partitioning."

Reviewer comment - correction No. 16: "Figure 13. Evolution of interface migration and interface migration velocity during the first and second partitioning. (two-stage Q-P process)."

Author Response

The response is shown in the response file.

Reviewer 2 Report

The authors showed carbon distribution and microstructural evolution during two-stage Q&P process of high strength steel by modelling. I think that they set up well the modelling conditions, and the modelling results were also quite interesting. However, experimental data such as microstructural analysis by SEM, TEM, XRD, and APT etc. is nessacery to support their modelling results. I recommend the authors to add experimental data into this paper to improve the quality of their paper.  

Author Response

(The authors gave the same response as above.)

Round 2

Reviewer 1 Report

Manuscript ID: materials-386485

(Carbon redistribution and microstructural evolution study during two-stage quenching and partitioning process of high strength steels by modeling), Materials journal

All necessary corrections have been successfully completed. It is my pleasure to give a recommendation for acceptance of the revised version (materials-386485-v2)

of paper entitled " Carbon redistribution and microstructural evolution study during two-stage quenching and partitioning process of high strength steels by modeling" for publication in Materials journal.

Reviewer 2 Report

I am pleased to inform that this article is ready for publication after careful revision.